# A Grey-Box Model of Laser Powder Directed Energy Deposition for Complex Scanning Strategy

**Mirna Poggi** [1], **Eleonora Atzeni** [1], **Michele De Chirico** [2] **and Alessandro Salmi** [1,*]

1    Department of Management and Production Engineering (DIGEP), Politecnico di Torino, Corso Duca degli Abruzzi 24, 10129 Torino, Italy; mirna.poggi@polito.it (M.P.); eleonora.atzeni@polito.it (E.A.)
2    Prima Additive S.r.l., Corso Re Umberto, 54, 10128 Torino, Italy; michele.dechirico@primaadditive.com
*     Correspondence: alessandro.salmi@polito.it; Tel.: +39-011-090-7263

**Abstract:** Directed Energy Deposition using a laser based system (DED-LB) is a technology that enables the repair of components, cutting costs and saving resources when it comes to valuable and expensive components. Furthermore, this method can be used in the production of multi-material components. Despite its benefits, DED-LB process has limitations as well, particularly in terms of resolution, surface quality, and dimensional accuracy. Optimisation of scanning parameters and strategies, as well as the use of new materials, appears to be advantageous in this regard. Simultaneously, the use of methods such as numerical simulation expedites the process of becoming familiar with the technology, thereby improving optimization tasks. DED-LB process starts with one track; the research and optimisation of its properties are crucial, as they affect the outcome of the DED-LB component. In this research article, a novel grey-box model that exhibits the ability to precisely predict the temperature distribution and track dimensions was introduced. The proposed model adopts a numerical–analytical methodology, yielding outcomes at a comparatively reduced computational expense while upholding precision in the obtained results. The proposed modelling approach is based on the solution of the heat equation coupled with an iterative feedback loop to quantify the power losses and ensure energy and mass balance at the melt pool. The model is used to forecast the temperature field and track characteristics for a collection of linear tracks while varying the main process parameters in order to study their effect on track characteristics. In addition, this model can be used to predict the course of more complex trajectories; to illustrate this, an application in which both circular and square tracks are made was presented.

**Keywords:** additive manufacturing; directed energy deposition; grey-box model; melt pool; scanning strategies

## 1. Introduction

Nowadays, the effort to reduce waste, material, and energy consumption plays a fundamental role in our society. From this perspective, the possibility of repairing and re-manufacturing high-performance and added-value components such as those used in the aeronautical, automotive, or aerospace fields offers significant advantages [1]. One of the Additive Manufacturing (AM) technologies that enables metal components to be repaired is Directed Energy Deposition (DED) [2]. DED is a category of AM processes that uses a high-energy source, such as a laser or an electron beam, to melt a feedstock material in powder or wire form directly where the material is needed. Among DED variants, laser-based directed energy deposition of powder material (DED-LB/Powder) is the most popular [3]. This technology utilises a fibre laser as the heat source and metal powder as the feedstock material. The useful laser power that reaches the substrate results in a local rise in temperature and the formation of the melt pool. Simultaneously, the powder is conveyed into the melt pool and rapidly melts, producing a raised track as a result of the added

material. While many studies have already demonstrated the ability and effectiveness of DED in restoring damaged components [4–6] or generating near-net-shaped parts [7], there are a number of open issues. The specific mechanism of material addition may induce high thermal gradients in the repaired part, which can affect the microstructure of the components, and the fast cooling rate generates residual stresses that can lead to distortion of the part [8,9]. These phenomena are governed by the numerous process parameters that define the final microstructure of the part and mechanical behaviour of the components.

Because of the large number of relevant physical phenomena and the complexity of their interactions, it is particularly difficult to link process parameters to quality of the final part. In order to enhance understanding of how process parameters affect outcomes while limiting experimental trials, numerical modelling and process simulations are widely used approaches [10]. Because the process ultimately turns out to be multi-scale, with effects ranging from microscopic to macroscopic, models may vary depending on the scale to which the input and output of simulations refer [11]. Several DED models in the literature focus on the microstructure of the part and its evolution during the process. Experiments remain the primary method of investigating microstructure; however, given the importance of the mechanical properties of the components, an increasing number of studies are placing greater emphasis on predicting and modelling the process of microstructure evolution [12,13]. One of the main inputs for these studies is the temperature field during the process, which can be derived experimentally or numerically using Finite Element Methods (FEM) or analytical solutions as needed [14]. On the other hand, if study of the deformation, stress, and temperature field is the primary concern, FEM techniques are among the most popular macro-scale models. FEMs are very advantageous in predicting the thermomechanical behaviour of even very large structures [15]. However, they are unable to estimate the size of the track and the amount of actual material deposited, which are necessary inputs for simulation. If the primary focus is on the properties of the tracks or the behaviour of the melt pool, meso-scale techniques are the ideal solution. Similarly, the thermal and physical behaviour of the melt pool is heavily influenced by process parameters such as the laser power, powder flow rate, and travel speed. Simulations using Computational Fluid Dynamics (CFD) can be utilised to assess the form and size of the melt pool and its adherence to the substrate [16]. This is a commonly employed methodology for the purpose of conducting track studies within the DED process as well as for various other AM technologies, such as the Powder Bed Fusion Laser Beam (PBF-LB) processes [17]. The objective of CFD models is to simulate the melt pool surface tension as a function of the temperature [18], which is essential for prediction of the melt pool dimensions. At the same time, CFD simulations involve high computational costs. In this instance, analytical models appear to be preferable, as it is feasible to predict the track geometry and the melt pool shape with reduced computational effort.

Both the aspect of the track and its measurements significantly impact the outcomes of each and every deposition, from the most tiny to the most substantial. A wide variety of numerical and analytical models have been developed and tested to investigate the impact of different process parameters on the track quality. One of the earliest models, developed by Colaço et al. [19], exploited the correlations between powder flow rate and track geometry, assumed to be circular in cross-section, at constant laser power and low dilution. Reflecting the predominance of surface tension forces, the model proposed by Pinkerton and Li [20] treats the melt pool boundaries as arcs of a circle rather than ellipses, and takes into consideration the elongation of the melt pool with increasing travel speed. The mass and power flow balances at the melt pool boundaries are used to estimate the melt pool and track geometry. To determine the shape of the melt pool, Picasso et al. [21] proposed a numerically solved three-dimensional geometric model of single-layer cladding. Because the depth of the melt pool is a mandatory input in their model, the authors projected that it would be in the shape of an ellipse and utilised the height of the cladding as the basis for this estimate. Alternatively, the theoretical models of single-layer cladding by Kaplan et al. [22] is based on decoupled mass and power balances, and does not include



this discrepancy. The authors employed the most well-known approach of measuring the size of the melt pool, which involves decoupling the heat and mass flow and pinpointing the spot at which the substrate solidus temperature is attained in one, two, or three dimensions. Furthermore, Pyre et al. [23] predicted the thermal field and track deposition geometry through a combined analytical–numerical model. First, the powder temperature was determined using analytic techniques, then the deposition geometry was predicted using a combination of analytical and numerical methods. The analytical temperature distribution owing to a moving heat source with distinct heat source models has been calculated for a wide range of conditions in both semi-infinite and finite three-dimensional domains, as evidenced by the literature. Due to the interconnected nature of the laser deposition processes, sequential solutions to the energy–mass balance equations are undesirable. In order to solve the DED process energy–mass balance equations, the model provided by Ahsan and Pinkerton [24] employs a linked analytical–numerical method. Model outputs, such as the temperature distribution in and around the melt pool, can be used to estimate important parameters such as the width, height, and depth of the deposition tracks. A significant shortcoming of this model is that it is only relevant to the theoretical straight path, as it does not include time as a variable and does not account for the transitory temperature field.

Hence, the present study has undertaken the task of generalising the Ahsan and Pinkerton [24] case by incorporating temporal dynamics into the framework, thereby making it adaptable to different scenarios that encompass the complex scanning strategies often seen in the real process. The model presented herein is an analytical model; as such, and in contrast to alternative methodologies, it possesses the inherent advantages of providing fundamental insights into the temperature evolution during the DED-LB/Powder process and the dimensions of the track with high reliability and expediency. Moreover, this particular model is executed within a numerical iteration loop wherein the intricate physical phenomena associated with the interaction between mass and energy are duly considered. In each iteration, the energy and mass balance are iteratively recalculated until the solution reaches a state of convergence. This study presents a simplified model that incorporates strong assumptions, such as the utilisation of constant material properties. It is worth noting that this model underwent rigorous verification and testing on a diverse range of rectilinear and non-rectilinear geometries. The outcomes of these experiments demonstrated that the model is capable of yielding precise results within a significantly reduced computational timeframe. This methodology can prove highly advantageous in the context of process parameter optimisation, as it enables researchers to effectively limit the range of variables that need to be investigated. Consequently, this approach facilitates the expedited identification of optimal solutions, a crucial attribute considering the rapidity with which industrial requirements evolve.

## 2. Modelling Approach

The proposed grey-box model primarily consists of a loop cycle (numerical part) in which the heat conduction equation (analytical part) is solved iteratively whenever the input power is adjusted. The implementation of the model is based on the following assumptions. The high conductivity of metals makes the thermal diffusion process the primary mechanism of metal deposition; therefore, the model is based on the analytical solution of the heat equation. The necessary input parameters can be grouped into the four categories presented in Table 1. The substrate and the deposition powder are of the same material, and are treated as homogeneous and isotropic. Moreover, in the Gaussian concentration distribution model of the powder flow rate, any powder that hits the melt pool is taken in by it and any other powder is deemed lost. Finally, the thermophysical properties are assumed to be constant, as in the previous work of Ahsan and Pinkerton [24]. As this is a strong assumption, a temperature that is in the middle of the range between room temperature and the melting temperature of the material should be used in order to minimise errors [24].

**Table 1.** Input parameters.

| Input Category | Input |
| --- | --- |
| Material properties | Thermal characteristics |
| | Mechanical characteristics |
| | Powder radius |
| Boundary conditions | Ambient and initial temperature |
| | Convection coefficient |
| Process parameters | Laser power |
| | Travel speed |
| | Powder flow rate |
| Machine parameters | Laser spot diameter |
| | Powder stream radius |

Figure 1 depicts the logic cycle that the suggested model uses, in which the initial step is to compute the useful power $P_u$, defined as follows:

$$P_u = \beta \cdot P - P_{losses} \tag{1}$$

where $\beta$ represents the substrate absorptivity [25], $P$ is the laser power, and $P_{losses}$ includes the losses due to radiation, convection, evaporation, and mass addition.

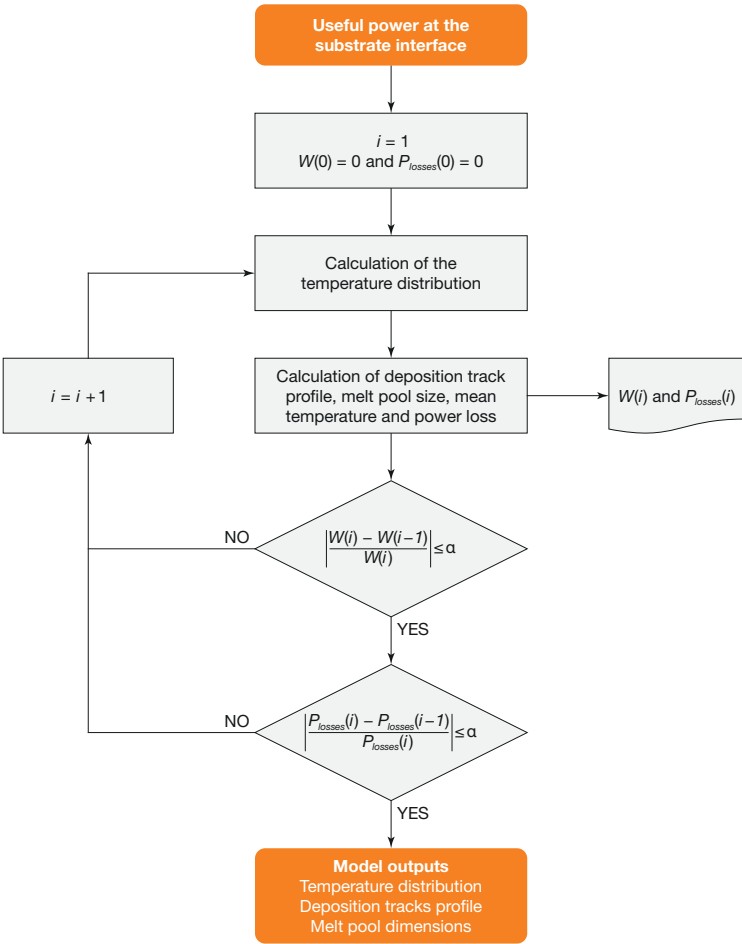

**Figure 1.** Schematic diagram of the flow chart for coupled deposition model.

The initial iteration involves setting the power losses to zero and evaluating the temperature by solving the heat conduction equation for each time step. Then, it is feasible to compute the size of the melt pool and the characteristics of the tracks. Without power losses, the resulting temperature field and melt pool size is overestimated. Then, it is possible to

calculate the first prediction of power losses owing to radiation, convection, evaporation, and mass addition using temperature and melt pool information. Afterward, the losses are input into Equation (1) and the loop recommences, solving the conduction equation with the revised temperature distribution while recalculating the melt pool characteristics and the power losses. The method is continued until the difference between the computation of the melt pool width $W$ and power losses $P_{losses}$ in the $n$-th iteration and the preceding iteration is equal to or less than the predetermined error $\alpha$.

The following section explores all iterative cycle steps from a mathematically analytical perspective.

### 2.1. Problem and Solution Formulation

The useful power consists of the energy required to melt the substrate and the powder in order to build the deposition track. Modelling of the heat source is essential, as it determines the thermal field inside the component. A Gaussian distribution was chosen to model the heat flux $\Phi$ for this investigation [8,26–30], as this distribution more closely resembles the laser beam energy distribution in the DED-LB process. The heat flux $\Phi$ can be represented utilising the useful power $P_u$ calculated with Equation (1) via the following expression:

$$\Phi(\mathbf{x}) = \frac{P_u}{2\pi r^2} \exp\left[-\frac{(x - x_c)^2 + (y - y_c)^2}{2r^2}\right] \cdot \frac{f(z)}{\mu} \tag{2}$$

where $\mathbf{x}$ is the vector of coordinate $(x, y, z)$, $r$ is the beam radius, $\mu$ is the absorption depth, $f(z)$ is a logic function which assumes values of $f(z) = 1$ for $-\mu < z < 0$ and $f(z) = 0$ for $z < -\mu$, and $x_c$ and $y_c$ are the coordinates of the centre of the beam spot, as reported in Figure 2.

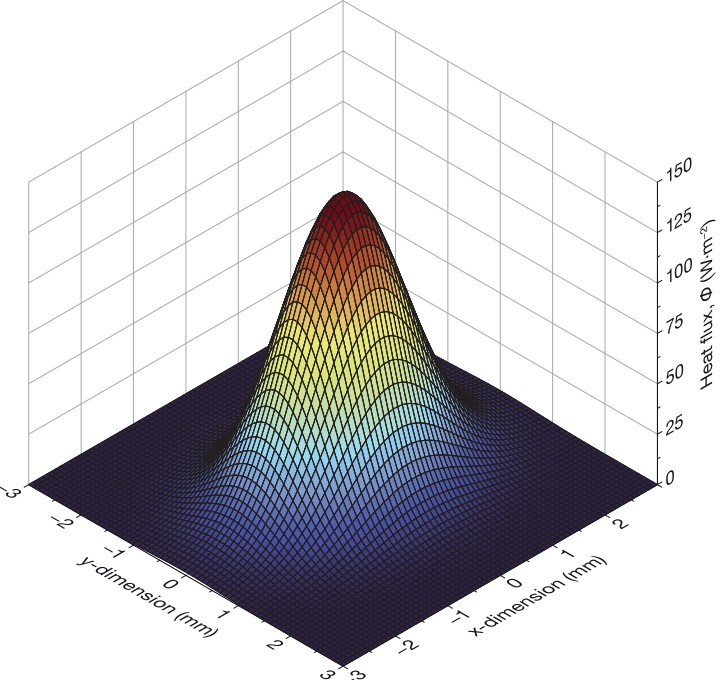

**Figure 2.** Gaussian distribution of heat flux with $P$ = 900 W and $r$ = 1 mm.

To accurately represent the temperature distribution in the substrate, it is essential to consider the substrate as semi-infinite when applying the heat conduction equation. This assumption is plausible given that the heat source is localised and considerably smaller than

the size of the substrate [24]. Therefore, the temperature distribution $T(\mathbf{x}, t)$ at any point of the domain $\mathbb{R}^2 \times \mathbb{R}_-$ and at any given time $t$ obeys the heat conduction equation [31]:

$$\frac{\partial T}{\partial t} - \kappa \nabla^2 T = \frac{\Phi}{\rho c_p} \tag{3}$$

where $\kappa = \lambda/(\rho c_p)$ is the thermal diffusivity, $\lambda$ is the thermal conductivity, $\rho$ is the material density, and $c_p$ is the specific heat capacity. As presented by Cline and Anthony [32], the solution of Equation (3) is obtained via superimposition of a series of point heat source solutions. Thus, applying the heat flux $\Phi$ at a generic point $(\xi, \eta, \zeta)$ on the surface influences the temperature at all points $(x, y, z)$ in the substrate at time $t$, and can be calculated through the convolution integral as follows:

$$T(\mathbf{x}, t) = T_{init} + \int_0^t \int_{\mathbb{R}^3} \frac{\Phi(\xi, \eta, \zeta, s)}{\rho c_p} \times \mathcal{G}(\xi, \eta, \zeta, s | x, y, z, t) \, d\xi \, d\eta \, d\zeta \, ds \tag{4}$$

where $T_{init}$ is the initial and uniform temperature of the substrate and $\mathcal{G}$ is the Green's function for the diffusion equation at the surface [33]:

$$\mathcal{G}(\mathbf{x}, t) = \left( \frac{1}{4\pi\kappa t} \right)^{3/2} \exp\left( -\frac{|\mathbf{x}|}{4\kappa t} \right) \tag{5}$$

In the case of given time-independent parameters, such as fixed power $P$ and travel speed $v$ during the deposition of a single track along the x-axis direction with a uniform initial temperature, the solution is a superimposition of the Gaussian heat distributions at the generic earlier time step $s$ when the laser was at the previous generic point of the substrate surface $(\xi, \eta, \zeta)$. Thus, the temperature distribution can be calculated through [32]:

$$T(\mathbf{x}) = T_{init} + \frac{P_u}{\rho c_p} \int_0^\infty \frac{\exp\left[ -\frac{(x - x_c + vs)^2 + (y - y_c)^2}{(2r^2 + 4\kappa s)} - \frac{z^2}{4\kappa s} \right]}{(\pi^3 \kappa s)^{1/2} (2r^2 + 4\kappa s)} \, ds \tag{6}$$

Nevertheless, as the real application of the DED-LB process necessitates a nonlinear scanning strategy and effective regulation of the deposition process, it is imperative that the model is able to modify the parameters as a function of time rather than assuming them as a constant. For these reasons, a generalisation of the model in [24] is carried out. The solution of Equation (3) can be written as the sum of two distinct contributions; the first contribution $T^I(\mathbf{x}, t)$ arises from the initial temperature, while the second contribution $T^\Phi(\mathbf{x}, t)$ originates from the heat source [34], as follows:

$$T(\mathbf{x}, t) = T^I(\mathbf{x}, t) + T^\Phi(\mathbf{x}, t). \tag{7}$$

Moreover, a uniform initial temperature is not required by the proposed method. Summation makes it simple to incorporate a time-dependent temperature distribution that, for instance, replicates the temperature history of a previously melted zone. Considering the partition described by Forslund et al. [34], where the laser path can be divided into $N$ segments with piecewise time-independent parameters, the temperature distribution can be expressed as follows:

$$T_n(\mathbf{x}, t) = T_{init} + T_1^\Phi(\mathbf{x}, t) + \sum_{k=1}^{n-1} T_{n,k}^I(\mathbf{x}, t) + T_n^\Phi(\mathbf{x}, t) \qquad for \ n = 2, 3, \cdots, N \tag{8}$$

where $T_{n,k}^I$ and $T_n^\Phi$ can be calculated through the following dimensionless integral:

$$T_{n,k}^I(\bar{\mathbf{x}}_k, \bar{t}_k) = \frac{P}{\sqrt{2}\pi^{3/2}\lambda r} \int_0^{\bar{t}_k^f} \frac{\bar{\tau}(\bar{\tau}^2 + \bar{t}_k^2)^{-1/2}}{(1 + \bar{\tau}^2 + \bar{t}_k^2)} \exp\left[ -\frac{(\bar{x}_k - \bar{x}_c + \bar{v}_{k,x}\bar{\tau}^2)^2 + (\bar{y}_k - \bar{y}_c + \bar{v}_{k,y}\bar{\tau}^2)^2}{1 + \bar{\tau}^2 + \bar{t}_k^2} - \frac{\bar{z}_k^2}{\bar{\tau}^2 + \bar{t}_k^2} \right] d\bar{\tau} \tag{9}$$

$$T_n^{\Phi}(\bar{\mathbf{x}}_n, \bar{t}_n) = \frac{P}{\sqrt{2}\pi^{3/2}\lambda r} \int_0^{\bar{t}_n} \frac{1}{1+\bar{s}^2} \exp\left[-\frac{(\bar{x}_n - \bar{x}_c + \bar{v}_{n,x}\bar{s}^2)^2 + (\bar{y}_n - \bar{y}_c + \bar{v}_{n,y}\bar{s}^2)^2}{1+\bar{s}^2} - \frac{\bar{z}^2}{\bar{s}^2}\right] d\bar{s} \tag{10}$$

Integrals are easier to compute in dimensionless form; in fact, the overline marks the dimensionless variables. The methodology for the calculation of these integrals can be found in [34].

### 2.1.1. Melt Pool Geometry

With the temperature of the substrate determined, it becomes feasible to evaluate the dimensions of the melt pool by analysing the isotherms in all three directions at the melting temperature $T_{melt}$. This methodology is further discussed in Section 3.1, which is dedicated to the implementation of the proposed model. The deposition track is formed by the powder dropping into the melt pool; hence, a detailed description of the melt pool geometry is essential. In order to define the melt pool geometry, it is necessary to introduce a new orthogonal reference frame $\mathfrak{R}'(O', x', y', z')$ that is integral with the heat source. The origin $O'(x'_0, y'_0, z'_0)$ of $\mathfrak{R}'$ coincides with the center of the heat source, which is characterized by the coordinates $(x_c, y_c, z_c)$ in the substrate reference frame $\mathfrak{R}(O, x, y, z)$. The typical shape of the melt pool can be approximated as two quarter-ellipsoids with respect to the laser direction. In Figure 3, both reference frames and the melt pool geometry are represented. Denoting the melt pool width as $W$, the front length as $L_f$, the rear length as $L_r$, and the depth as $D$, the melt pool geometry can be represented by the Equation (11), where the x'-axis is coincident with the direction of the laser and positive in the direction of the movement:

$$\begin{cases} \dfrac{x'^2}{L_f^2} + \dfrac{4y'^2}{W^2} + \dfrac{z'^2}{D^2} = 1 & \text{if } x' \geq 0, z' \leq 0 \\[3mm] \dfrac{x'^2}{L_r^2} + \dfrac{4y'^2}{W^2} + \dfrac{z'^2}{D^2} = 1 & \text{if } x' < 0, z' \leq 0 \end{cases} \tag{11}$$

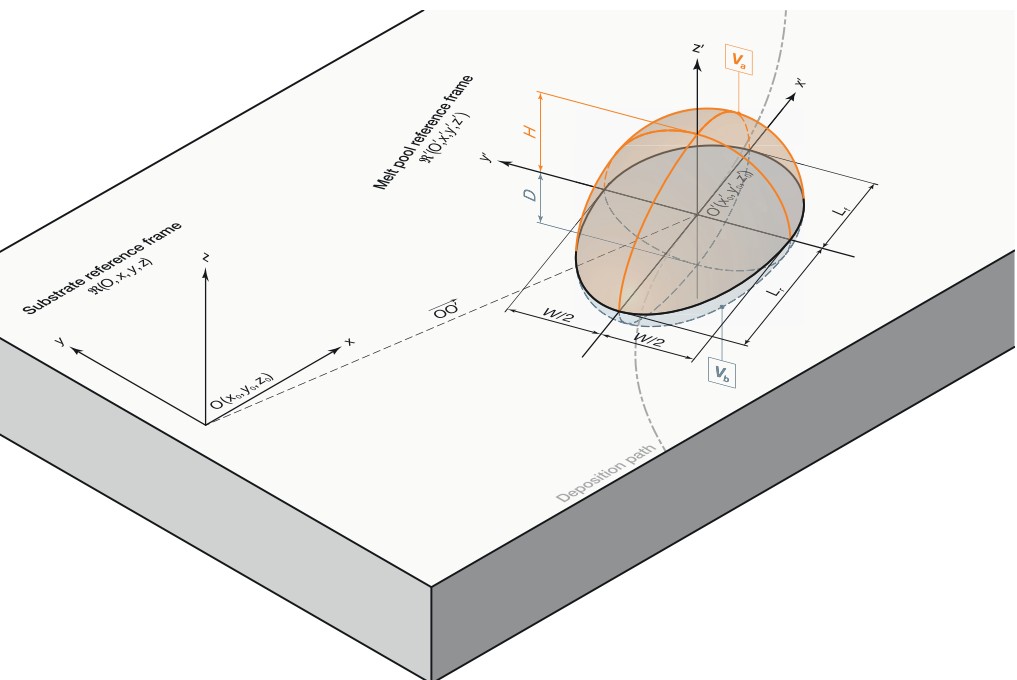

**Figure 3.** Substrate reference frame $\mathfrak{R}$, melt pool reference frame $\mathfrak{R}'$ and melt pool geometry.

In order to calculate the deposition height $H$, first of all, the powder mass flux $q_{pf}$ must be derived from the powder flow rate $Q_p$ delivered by the deposition head through the following equation:

$$q_{pf}(x', y') = \frac{2Q_p}{\pi r_{pf}^2} \exp\left(-2\frac{x'^2 + y'^2}{r_{pf}^2}\right) \tag{12}$$

Then, the deposition height $H$ of the track can be derived as the maximum value of the function of the deposited material $H'(x', y')$ calculated by integrating the Gaussian powder mass flux over the region of the melt pool, as follows:

$$H'(x', y') = \frac{1}{\rho v} \int_{x_r'(y')}^{x_f'(y')} q_{pf} \, dx' \tag{13}$$

$$H = \max\{H'(x', y')\} \tag{14}$$

where the integration limits $x_f'$ and $x_r'$ of Equation (13) are respectively the front and rear limits of the melt pool at the surface $z = 0$. These can be calculated by the following equations:

$$x_f'(y') = L_f \sqrt{1 - \frac{4y'^2}{W^2}} \tag{15}$$

$$x_r'(y') = -L_r \sqrt{1 - \frac{4y'^2}{W^2}} \tag{16}$$

All information regarding the height and width of the melt pool is necessary for calculating the power losses, which is addressed in the next section.

### 2.1.2. Power Loss Calculation

The power losses related to the process can be categorized as evaporation, convection, radiation, and sustained mass addition [35]. To calculate the evaporation, convection, and radiation losses, the mean temperature of the melt pool must be evaluated. This can be done, as suggested by [24], by dividing the deposition track into two portions as shown in Figure 3. The volume $V_a$ above the top surface of the substrate and the volume $V_b$ below the top surface of the substrate, as it is a quasi-stationary state. The first volume is bound to the mass balance above the substrate surface and is assumed at the melting temperature $T_{melt}$. The volume $V_a$ can be derived by integrating the deposition height in both $x'$ and $y'$ directions:

$$V_a = \int_{-W/2}^{W/2} \int_{x_r'(y')}^{x_f'(y')} H'(x', y') \, dx' dy' \tag{17}$$

where the volume $V_b$ is bound to the energy balance and is assumed at the highest temperature $T_{peak}$. The portion below the substrate surface can be approximated as a semi-ellipsoid, and its volume results are as follows:

$$V_b = \frac{\pi}{3} \frac{W}{2} D\left(L_f + L_r\right) \tag{18}$$

Thus, the mean temperature $T_{mean}$ of the deposition track can be calculated through the enthalpy balance, as follows:

$$T_{mean} = \frac{V_a T_{melt} + V_b T_{peak}}{V_a + V_b} \tag{19}$$

In the previous study of Pinkerton et al. [36], the convective losses $P_{conv}$ and radiation losses $P_{rad}$ were calculated as follows:

$$P_{conv} = h\,(T_{mean} - T_a) \int_{x'_r(y')}^{x'_f(y')} H'(x', y')\, dx' \tag{20}$$

$$P_{rad} = \varepsilon\sigma\,(T_{mean}^4 - T_a^4) \int_{x'_r(y')}^{x'_f(y')} H'(x', y')\, dx' \tag{21}$$

where $h$ is the convection coefficient, $\varepsilon$ is the material emissivity, and $\sigma$ is the Stefan–Boltzman constant. For the evaporation losses $P_{evap}$, the formula presented by Salcudean et al. [37] was used:

$$P_{evap} = \left[ \frac{\exp(10.941 - 18,836 \cdot T_{mean}^{-1})}{\sqrt{T_{mean}}} \right] L_v \int_{x'_r(y')}^{x'_f(y')} H'\, dx' \tag{22}$$

where $L_v$ is the latent heat of vaporization. Finally, the addition of powder tends to reduce the temperature of the melt pool, meaning that its size is reduced as well. For this reason, the power $P_{melt}$ required to melt the powder must be counted as a loss and subtracted from the initial beam power. The profile of the track is used to calculate the mass of powder that falls in the melt pool:

$$P_{melt} = \rho\,v[c_p(T_{melt} - T_a) + L_m] \int_{-W/2}^{W/2} H'(x', y')\, dy' \tag{23}$$

where $L_m$ is the latent heat of fusion. In the end, the overall power losses $P_{losses}$ are calculated on the base of the melt pool geometry, and can be written as follows:

$$P_{losses} = P_{conv} + P_{rad} + P_{evap} + P_m. \tag{24}$$

## 3. Model Implementation and Validation

The analytical model described in the previous section was implemented in the simulation software MATLAB R2023b by MatWorks® (Natick, MA, USA) to calculate the temperature field and the melt pool geometry as a function of process parameters. It was then validated on experimental tests to check its consistency.

### 3.1. Model Implementation

The proposed model was implemented in MATLAB, as presented in Algorithm 1. First, the temperature field was computed by solving Equation (8) using the MATLAB quadgk function, which solves integrals with high-order global adaptive (Gauss–Kronrod) quadrature and default error tolerances. This information was then passed to the meltpoolFunction, which is designed to represent the shape of the melt pool and facilitate its visualization. In particular, the function assigns 1 to each temperature value over the melt temperature and 0 to the others. Next, the second-degree polynomial fit function was employed to determine those ellipsoid quarters that best represent the data previously gathered. Then, the other specific designed function MPfunction was used to derive the necessary information of the track such as the width and the front and rear lengths. Using this information, it is possible to compute the deposition height by solving Equation (13), as well as the average temperature and power loss using the methods described previously. The difference in melt pool width and power losses between the current iteration and the previous iteration can then be computed. If the difference is less than the error $\alpha$, that is set at one percent, the iterative cycle is terminated, and the computation concludes; otherwise, the cycle is restarted using the new power value as input and the power losses are calculated according to Equation (24).

**Algorithm 1:** Alghoritm of the logic loop cycle.

```
begin
i = 0
while test = 1 do
    T = temperatureFunction(P_u)
    T_max = max(T)
    MeltPoolT = meltpoolFunction(T)
    MPDimensions = MPfunction(MeltPoolT)
    H = formulaHeight(MPDimensions)
    T_mean = meanTemperatureFunction( T_max , MPDimensions)
    P_losses= formulaPower(T_mean , MPDimesnions)
    W(i) = functionWidth(MPDimensions)
    DeltaW = |(W(i) − W(i − 1))/W(i)|
    DeltaP =|(P_losses(i) − P_losses(i − 1))/P_losses(i)|
    i = i + 1
    if DeltaW ≤ 0.01 then
        if DeltaP ≤ 0.01 then
        |   test = 0
        else
        |   test = 1
        end
    else
    |   test = 1
    end
end
```

In the simulation values, the substrate absorbivity $\beta$ is set at 0.5 [3]. The selected material properties of AISI 316L are reported in Table 2.

**Table 2.** Properties of AISI 316L.

| Property | Symbol | Value | Units |
|---|---|---|---|
| Density | $\rho$ | $8 \times 10^{-6}$ | kg·mm$^{-3}$ |
| Specific heat capacity | $c_p$ | 800 | J·kg$^{-1}$·K$^{-1}$ |
| Thermal conductivity | $\lambda$ | $21.4 \times 10^{-3}$ | W·mm$^{-1}$·K$^{-1}$ |
| Thermal diffusivity | $\kappa$ | 3.34 | mm$^2$·s$^{-1}$ |
| Latent heat of fusion | $L_v$ | $260 \times 10^3$ | J·kg$^{-1}$ |
| Latent heat of vaporization | $L_m$ | 6259.5 | J·kg$^{-1}$ |
| Convective heat transfer | $h$ | $1 \times 10^{-3}$ | W·mm$^{-2}$·K$^{-1}$ |
| Stefan Bolzman constant | $\sigma$ | $5.67 \times 10^{-14}$ | W·mm$^{-2}$·K$^{-4}$ |
| Emissivity | $\varepsilon$ | 0.6 | − |

### 3.2. Experimental Validation of the Model

A set of experimental tests consisting of AISI 316L stainless steel 30 mm long linear tracks was used to validate the suggested method. This validation is necessary because the formulation of the problem is an analytical solution that necessitates the application of strong assumptions, and the phenomenon being represented is extremely complicated due to the simultaneous occurrence of numerous physical effects. The process parameters were selected following a review of the literature [38–41], and the corresponding values are reported in Table 3. After this initial validation, the model was tested on six square tests of 25 mm side length and six circular tests of 25 mm diameter to evaluate its feasibility. For this second series of samples, it was decided that three distinct parameter sets with the same energy density value would be used. This value was determined based on earlier research by the authors into the quality of deposition tracks on AISI 316L powder and

substrate [42], which showed that better tracks can be achieved with an energy density of 14.32 J·mm$^{-2}$. Two powder flow rate values were applied to each pair of parameters, as shown in Table 4. Samples were produced using the LASERDYNE$^{®}$ 430 system by Prima Additive (Torino, Italy). The DED-LB/Powder machine was provided with a fibre laser with a maximum power of 1 kW, a laser spot of 2 mm, and a coaxial deposition head with four nozzles. The powder used in the experimental investigation was MetcoAdd 316L-D by Oerlikon (Freienbach, Switzerland), consisting of gas-atomized pre-alloyed AISI 316L with particle-size distribution ranging from 45 µm to 160 µm. A Leica S9i (Wetzlar, Germany) optical microscope (OM) was used to capture images of each deposition. The images were analysed with ImageJ 1.52t software (Bethesda, MD, USA) to extract the width of the deposited tracks. ImageJ is capable of recognising the track outline and calculating its area; using this information and the track length, it is possible to calculate the average track width. In order to examine the cross-section of the deposited tracks, the substrates and tracks were cut using Wire Electrical Discharge Machining (W-EDM) after the initial examination. Hence, sample cross-sections were examined under the microscope once more and the height of each track was determined.

**Table 3.** Parameters for linear tracks.

| Track ID | Power (W) | Travel Speed (mm·min$^{-1}$) | Powder Flow Rate (g·min$^{-1}$) |
|---|---|---|---|
| L01 | 700 | 600 | 8.9 |
| L02 | 700 | 600 | 13.1 |
| L03 | 700 | 600 | 17.2 |
| L04 | 700 | 800 | 8.9 |
| L05 | 700 | 800 | 13.1 |
| L06 | 700 | 800 | 17.2 |
| L07 | 700 | 1000 | 8.9 |
| L08 | 700 | 1000 | 13.1 |
| L09 | 700 | 1000 | 17.2 |
| L10 | 900 | 600 | 8.9 |
| L11 | 900 | 600 | 13.1 |
| L12 | 900 | 600 | 17.2 |
| L13 | 900 | 800 | 8.9 |
| L14 | 900 | 800 | 13.1 |
| L15 | 900 | 800 | 17.2 |
| L16 | 900 | 1000 | 8.9 |
| L17 | 900 | 1000 | 13.1 |
| L18 | 900 | 1000 | 17.2 |

**Table 4.** Parameters for circular (Cxx) and squared (Sxx) tracks.

| Track ID | Power (W) | Travel Speed (mm·min$^{-1}$) | Powder Flow Rate (g·min$^{-1}$) |
|---|---|---|---|
| C01/S01 | 720 | 480 | 13.1 |
| C02/S02 | 720 | 480 | 17.2 |
| C03/S03 | 810 | 540 | 13.1 |
| C04/S04 | 810 | 540 | 17.2 |
| C05/S05 | 900 | 600 | 13.1 |
| C06/S06 | 900 | 600 | 17.2 |

## 4. Results and Discussion

The outcomes of our simulations and experiments are reported in this section. First, the simulated temperature distribution is presented, followed by a discussion of the results pertaining to the characteristics of the linear tracks, then the width and height of the circular and squared tracks.

### 4.1. Temperature Distribution

Figure 4 depicts the temperature distribution obtained by the proposed model for linear track L07. The maximum temperature reached is 1652 °C, and the isotherms of the melting temperature have the typical shape of two ellipses. The asymmetry in the melt pool is due to the laser beam motion and the heat conduction of the substrate. The temperature distribution obtained for all the parameters shows the same trend and typical shape. The peak temperature range varies from a minimum of 1565 °C for L09 linear track and a maximum of 2061 °C for L10 linear track. Obviously, this variation is due to the different values of power, travel speed, and powder flow rate, which imply a different amount of energy reaching the substrate, resulting in varying temperatures and melt pool sizes. The peak temperature results of for all the linear tracks are presented in Table 5.

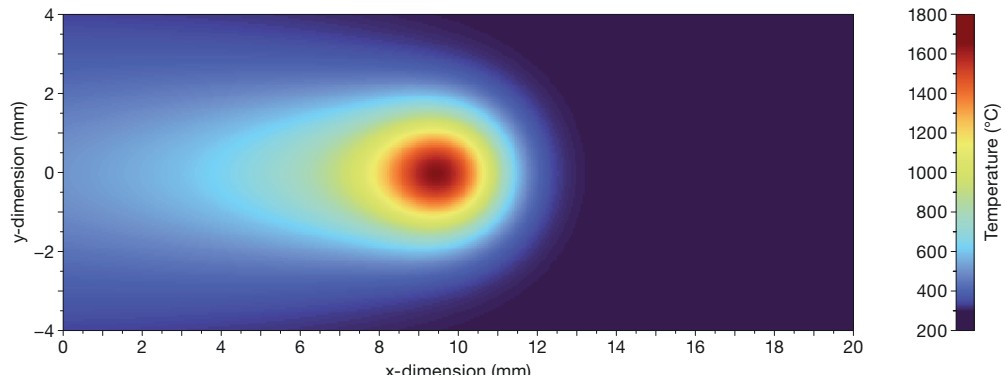

**Figure 4.** Temperature distribution of L07 linear track ($P$ = 900 W, $v$ = 1000 mm·min$^{-1}$, $Q_p$ = 8.9 g·min$^{-1}$) at time $t$ = 0.6 s.

**Table 5.** Peak temperature numerical results for linear tracks.

| Track ID | Peak Temperature (°C) |
|---|---|
| L01 | 1824 |
| L02 | 1763 |
| L03 | 1686 |
| L04 | 1730 |
| L05 | 1688 |
| L06 | 1590 |
| L07 | 1652 |
| L08 | 1624 |
| L09 | 1565 |
| L10 | 2061 |
| L11 | 1952 |
| L12 | 1817 |
| L13 | 1962 |
| L14 | 1881 |
| L15 | 1808 |
| L16 | 1934 |
| L17 | 1864 |
| L18 | 1796 |

In the case of circular and squared tracks, where the given energy density is the same, the temperature variance is significantly smaller. In both instances, however, the highest temperature is found in C05 and S05 and the lowest in C02 and S02. First, even if the energy density is the same, the amount of material provided is different, meaning that less energy reaches the substrate, as more energy is required to melt the material. Second, it is noticeable that the power has a significant effect on the increase in temperature; in C05

and S05, the power is 900 W, whereas in C02 and S02 the power is 720 W. The results are presented in Tables 6 and 7.

**Table 6.** Peak temperature numerical results for circular tracks.

| Track ID | Peak Temperature (°C) |
|---|---|
| C01 | 2174 |
| C02 | 2063 |
| C03 | 2144 |
| C04 | 2037 |
| C05 | 2278 |
| C06 | 2175 |

**Table 7.** Peak temperature numerical results for squared tracks.

| Track ID | Peak Temperature (°C) |
|---|---|
| S01 | 2268 |
| S02 | 2151 |
| S03 | 2229 |
| S04 | 2173 |
| S05 | 2383 |
| S06 | 2275 |

*4.2. Track Dimensions*

The matrix showing the temperature distribution on the plane $z = 0$ is fed into a code that determines the shape of the melt pool by assigning a value of 1 to places where the temperature is above the melting point and 0 to places where the temperature is below it. The outcomes for L09 and L10 linear tracks are reported in Figure 5, where the yellow represents the melt pool and the blue represents the non-melted substrate. As depicted in the picture, the melt pool has the conventional droplet shape and the expected two-elliptical shape predicted by the model. Beam motion causes the characteristic asymmetry, and its tail expands as the travel speed increases.

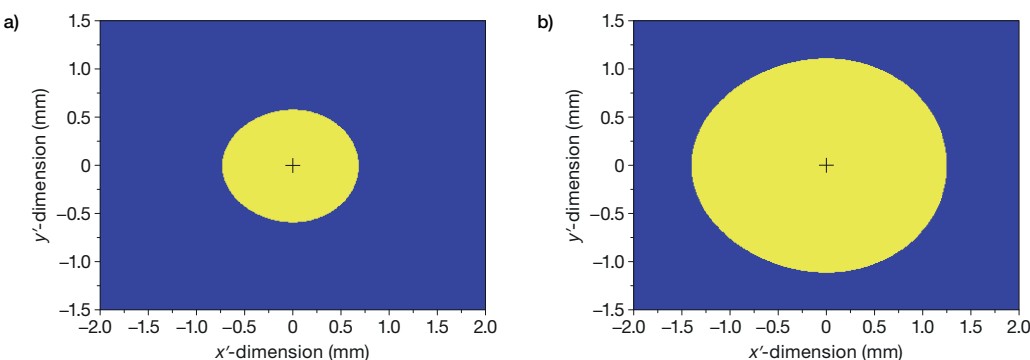

**Figure 5.** Melt pool representation: (**a**) L09 linear track ($P = 700$ W, $v = 1000$ mm·min$^{-1}$, $Q_p = 17.2$ g·min$^{-1}$) and (**b**) L10 linear track ($P = 900$ W, $v = 600$ mm·min$^{-1}$, $Q_p = 8.9$ g·min$^{-1}$).

Next, the eighteen linear tracks were evaluated in order to measure their widths and heights. The top view and cross-section pictures of the linear tracks obtained via OM are shown in Figure 6. Moreover, the experimental values together with the related confidence intervals and the simulation results in terms of width and height are listed in Table 8. The typical inaccuracy for the model is approximately 8% for width and 12% for deposition height. The three tests run at 700 W with the maximum powder flow rate of 17.2 g·min$^{-1}$ each produced the results with the most significant error in terms of width. In each of these

three scenarios, the model had a propensity to provide a lower estimate than the actual track width. Because of this, it is clear that the power losses have been overestimated; as a result, the simulation produces a value that is lower than the actual width. The primary reason for the discrepancy between the simulation and the model is that the simulation does not consider a different set of dynamics, which is due to the inclusion of additional physical phenomena in the process. Furthermore, the strong approximation used in the presentation of the analytical formulation of the problem may be the cause of discrepancies between the model and the sample test. Additionally, there is a risk associated with the experimental test, as a variety of different phenomena could interfere with it. In any case, the experimental tests show a similar trend with respect to the model; in light of these findings, it is possible to use the model in a procedure that employs a sophisticated scanning method.

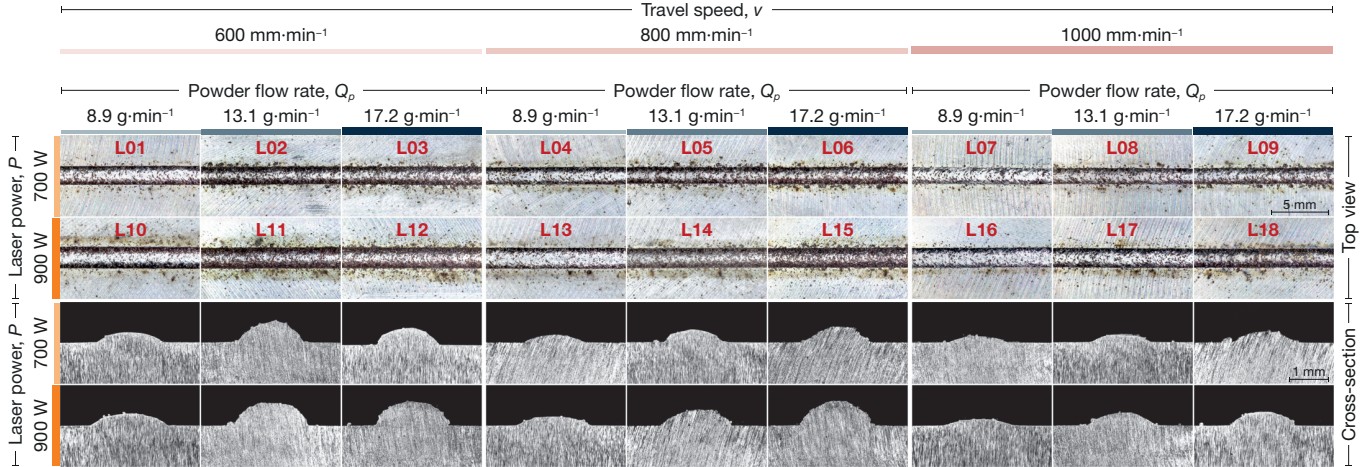

**Figure 6.** Top view and cross-section view of linear tracks.

**Table 8.** Experimental measurements and numerical results for each single linear track.

| Track ID | $W_{exp}$ (mm) | $W_{num}$ (mm) | $\Delta W$ (%) | $H_{exp}$ (mm) | $H_{num}$ (mm) | $\Delta H$ (%) |
|---|---|---|---|---|---|---|
| L01 | $1.67 \pm 0.02$ | 1.80 | 8% | $0.25 \pm 0.02$ | 0.31 | 26% |
| L02 | $1.62 \pm 0.05$ | 1.67 | 3% | $0.50 \pm 0.01$ | 0.43 | $-15\%$ |
| L03 | $1.62 \pm 0.07$ | 1.38 | $-15\%$ | $0.42 \pm 0.01$ | 0.47 | 12% |
| L04 | $1.55 \pm 0.04$ | 1.56 | 1% | $0.23 \pm 0.01$ | 0.21 | $-11\%$ |
| L05 | $1.51 \pm 0.03$ | 1.46 | $-3\%$ | $0.33 \pm 0.01$ | 0.28 | $-13\%$ |
| L06 | $1.61 \pm 0.07$ | 1.29 | $-20\%$ | $0.35 \pm 0.01$ | 0.33 | $-5\%$ |
| L07 | $1.45 \pm 0.04$ | 1.34 | $-7\%$ | $0.17 \pm 0.01$ | 0.15 | $-15\%$ |
| L08 | $1.43 \pm 0.02$ | 1.26 | $-12\%$ | $0.21 \pm 0.01$ | 0.20 | $-6\%$ |
| L09 | $1.49 \pm 0.03$ | 1.19 | $-20\%$ | $0.21 \pm 0.02$ | 0.20 | $-5\%$ |
| L10 | $2.00 \pm 0.02$ | 2.22 | 11% | $0.36 \pm 0.02$ | 0.38 | 4% |
| L11 | $1.88 \pm 0.06$ | 2.09 | 11% | $0.56 \pm 0.02$ | 0.54 | $-3\%$ |
| L12 | $1.86 \pm 0.04$ | 1.95 | 5% | $0.58 \pm 0.02$ | 0.66 | 13% |
| L13 | $1.90 \pm 0.02$ | 1.99 | 5% | $0.30 \pm 0.01$ | 0.27 | $-10\%$ |
| L14 | $1.79 \pm 0.04$ | 1.87 | 4% | $0.34 \pm 0.01$ | 0.37 | 9% |
| L15 | $1.78 \pm 0.05$ | 1.77 | $-1\%$ | $0.62 \pm 0.03$ | 0.46 | $-25\%$ |
| L16 | $1.81 \pm 0.03$ | 1.93 | 7% | $0.17 \pm 0.01$ | 0.22 | 24% |
| L17 | $1.75 \pm 0.04$ | 1.86 | 7% | $0.33 \pm 0.01$ | 0.30 | $-7\%$ |
| L18 | $1.68 \pm 0.05$ | 1.75 | 4% | $0.33 \pm 0.01$ | 0.37 | 12% |

Figures 7 and 8 display the top view and cross-section view of the samples with circular and squared trajectories. The tracks were analyzed in the same way as before. Tables 9 and 10 present the experimental values together with the related confidence intervals and simulation results for the circular tracks and the squared tracks, respectively. Regarding the

average simulation error of the track width, the values of 3% for circular tracks and 2% for the squared tracks are extremely minimal. In comparison, the average inaccuracy for the deposition height is 12% in both instances. The average error for the tracks is smaller in this second series of simulations than in the first, likely owing to the use of a constant energy density value in the second set, resulting in a more accurate absorptivity coefficient. In fact, in the preceding case, the highest fluctuation occurred at a lower energy density level, specifically at lower powers and a larger powder flow rate. The same is observed in the circular track simulations, where a larger powder flow rate results in the greatest inaccuracy. Regarding the mean error of the tracks, the first and second sets of testing yielded identical results. In the second series of testing, with the circular and squared tracks, the model showed a tendency to overestimate the height of the track. This is likely owing to the fact that the deposition height is computed using an analytical formula based on the size of the melt pool. Nevertheless, for non-rectilinear trajectories the inertia component of the powder that tends to retain direction is stronger; thus, less powder reaches the melt pool.

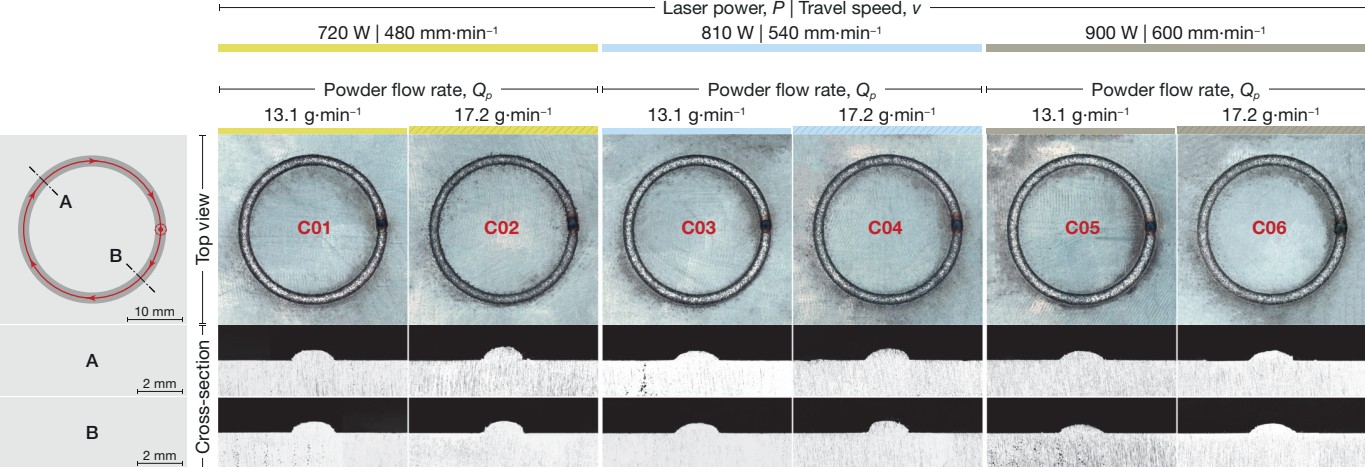

**Figure 7.** Top view and A and B cross-section views of circular tracks.

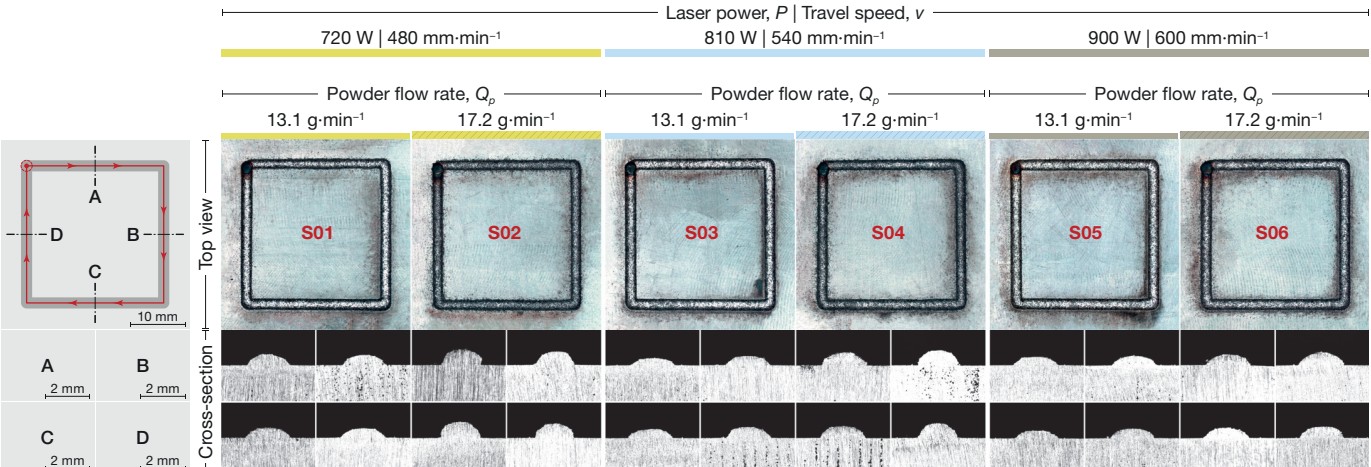

**Figure 8.** Top view and A, B, C, and D cross-section views of squared tracks.

**Table 9.** Experimental measurements and numerical results for each single circular track.

| Track ID | $W_{exp}$ (mm) | $W_{num}$ (mm) | $\Delta W$ (%) | $H_{exp}$ (mm) | $H_{num}$ (mm) | $\Delta H$ (%) |
|---|---|---|---|---|---|---|
| C01 | $1.91 \pm 0.03$ | 1.93 | 1% | $0.48 \pm 0.01$ | 0.52 | 10% |
| C02 | $1.77 \pm 0.04$ | 1.70 | $-4$% | $0.56 \pm 0.01$ | 0.61 | 8% |
| C03 | $1.99 \pm 0.01$ | 2.00 | 1% | $0.43 \pm 0.02$ | 0.48 | 10% |
| C04 | $1.88 \pm 0.07$ | 1.81 | $-4$% | $0.52 \pm 0.01$ | 0.57 | 10% |
| C05 | $2.05 \pm 0.04$ | 2.10 | 2% | $0.39 \pm 0.01$ | 0.45 | 16% |
| C06 | $2.00 \pm 0.02$ | 1.92 | $-4$% | $0.44 \pm 0.02$ | 0.54 | 22% |

**Table 10.** Experimental measurements and numerical results for each single squared track.

| Track ID | $W_{exp}$ (mm) | $W_{num}$ (mm) | $\Delta W$ (%) | $H_{exp}$ (mm) | $H_{num}$ (mm) | $\Delta H$ (%) |
|---|---|---|---|---|---|---|
| S01 | $1.92 \pm 0.03$ | 1.95 | 2% | $0.46 \pm 0.03$ | 0.53 | 22% |
| S02 | $1.70 \pm 0.04$ | 1.71 | 1% | $0.67 \pm 0.06$ | 0.61 | $-8$% |
| S03 | $1.99 \pm 0.01$ | 2.00 | +0% | $0.41 \pm 0.05$ | 0.49 | 19% |
| S04 | $1.83 \pm 0.07$ | 1.78 | $-3$% | $0.61 \pm 0.06$ | 0.56 | $-7$% |
| S05 | $2.02 \pm 0.04$ | 2.12 | 5% | $0.40 \pm 0.03$ | 0.45 | 13% |
| S06 | $1.98 \pm 0.02$ | 1.91 | -4% | $0.51 \pm 0.04$ | 0.55 | 9% |

*4.3. Discussion*

Deposition track geometry is a significant element in determining the final dimensions and surface morphology of the manufactured part, constituting one of the most essential characteristics of the DED process. The graphs in Figure 9 show the width and height of linear track depositions for different level of laser power and travel speed at a constant powder flow rate. In addition, the results of the experimental tests and their comparison with the results of the model are presented in graph form. The graphs demonstrate that the width of the track is largely reliant on the energy available at the substrate. The energy that strikes the substrate depends on several variables, including the powder flow rate, travel speed, and laser power. Clearly, as the laser power grows, the available energy and the width of the track grow as well. As the amount of power increases, the losses due to evaporation, radiation, and convection all increase. Due to these factors, the increase in track width is not proportional to the increase in power. On the other hand, Figure 9 shows that the increment of the travel speed leads to a decrease in track width. The same effect is valid for the deposition height, as the increase in travel speed means that a lower energy density is available to melt the incoming powder into the substrate. Moreover, the deposition height is not affected by laser power at low powder flow rates. The effect is much more pronounced at higher flow rates. This means that a share of the power is not used for a low value of the powder flow rate, as there is insufficient incoming powder to be melted. The resulting "excess" of power affects the substrate, increasing the heat-affected zone or the track width. In fact, our results show a greater difference in width values at different power levels for the same lower powder flow rates.

Moreover, it can also be observed that the width of the melt pool decreases as the powder flow rate increases due to the decrease in temperature at the substrate surface. This is attributed to the fact that a greater powder flow rate means a greater laser beam attenuation and higher power being required to sustain mass addition. This leads to less energy being available and a decrease in the melt pool width. Moreover, increasing the travel speed has the same effect as increasing the powder flow rate. This is due to the lower energetic delivery for the minor time that the laser stays on the substrate, and makes for higher cooling rates. Regarding the deposition height, it is mainly dependent on the powder flow rate, as shown in the graphs in Figure 9. In fact, at a constant travel speed, increasing the powder flow rate determines an increase in height, which is opposite to the trend for width. Conversely, with regard to the width, an increase in travel speed

results in a reduction in the deposition height. This happens for the same reasons, that is, because the laser and the deposition head spend less time at the same point on the substrate. Furthermore, increasing the travel speed reduces the width of the melt pool, meaning that less powder has a chance of being captured and included in the deposition track. Only a slight increase in deposition height is encountered with laser beam power increase; in fact, there is a very small difference between the heights obtained for 900 W power and 700 W.

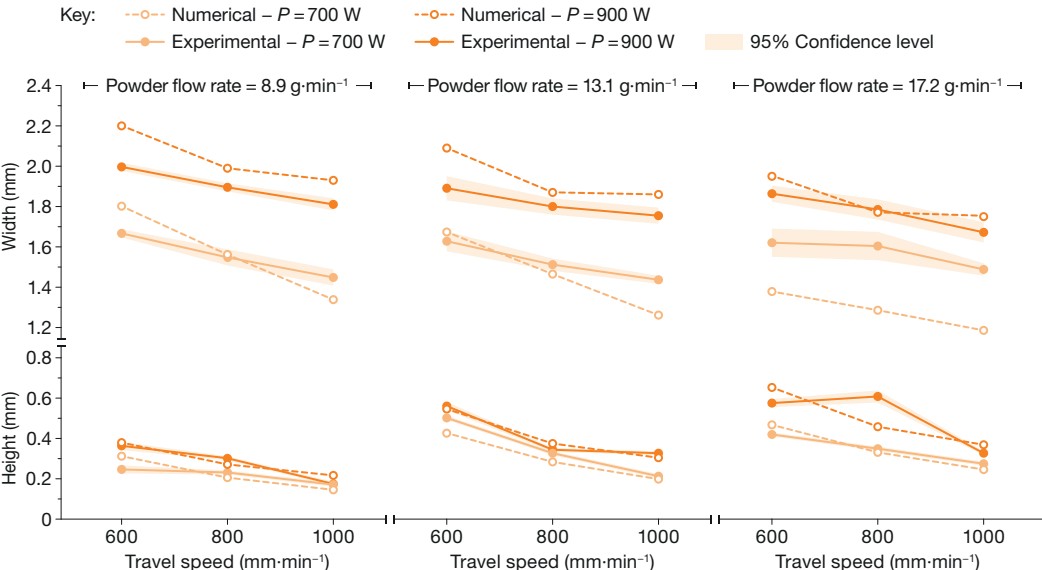

**Figure 9.** Widths and heights of the tracks as predicted by the grey-box model in comparison to experimental data for the linear tracks.

The graphs depicted in Figures 10 and 11 represent the experimental and simulation results of the width and height for circular and squared tracks, respectively. In all, cases the substrate and powder are affected by the same amount of energy, with the values of power and travel speed varying simultaneously. In previous studies [43], it has been shown that the track size is reduced as the power decreases, and that conversely, the track grows as the travel speed decreases. In this case, both travel speed and power decrease simultaneously, and it can be seen from the graphs in Figures 10 and 11 that the track decreases as well. This implies that the power has a greater effect on track size than the travel speed. The same is evident from the linear tracks, where the differences between tracks with different velocities and the same power are smaller than the differences between tracks with the same travel speed and different power. This is due to the fact that the power is what actually delivers the energy necessary to melt the material and powder, while the travel speed modifies the amount of time for which the energy is delivered.

In addition, with regard to the powder flow rate, it can be observed that the tracks become smaller as the amount of powder being supplied increases. This is because power losses increase with the amount of mass needing to be heated and melted. For this reason, raising the powder flow rate to maintain the same track width necessitates additional power. Furthermore, it can be observed that the difference in power is less obvious with lower values of powder flow rate, whereas it is more apparent at greater values of powder flow rate. This implies that power plays a more crucial role when increasing the mass.

Finally, under the same process settings, the difference between the circular and squared depositions in terms of track width is not as pronounced. As the circular enclosing area is smaller than the squared track, there is less heat loss through conduction; thus, slightly greater values are found for the circular tracks. This phenomenon can be attributed to the reduced quantity of material injected into the melting pool. The decreased deposition height of circular tracks is due to the inertia of the powder for the curved geometry.

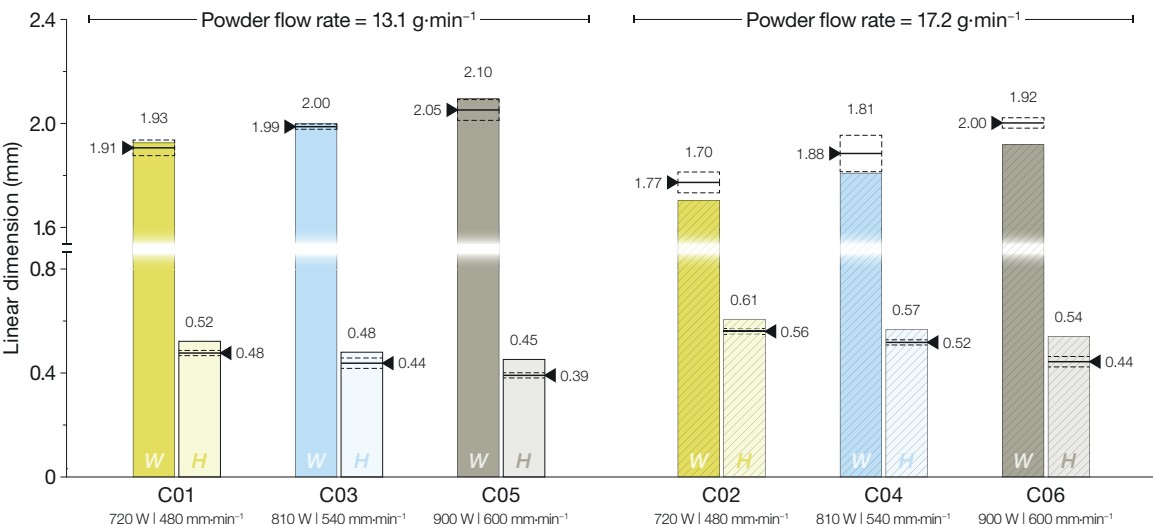

**Figure 10.** Widths and heights of the tracks as predicted by the grey-box model in comparison with the experimental data for circular tracks.

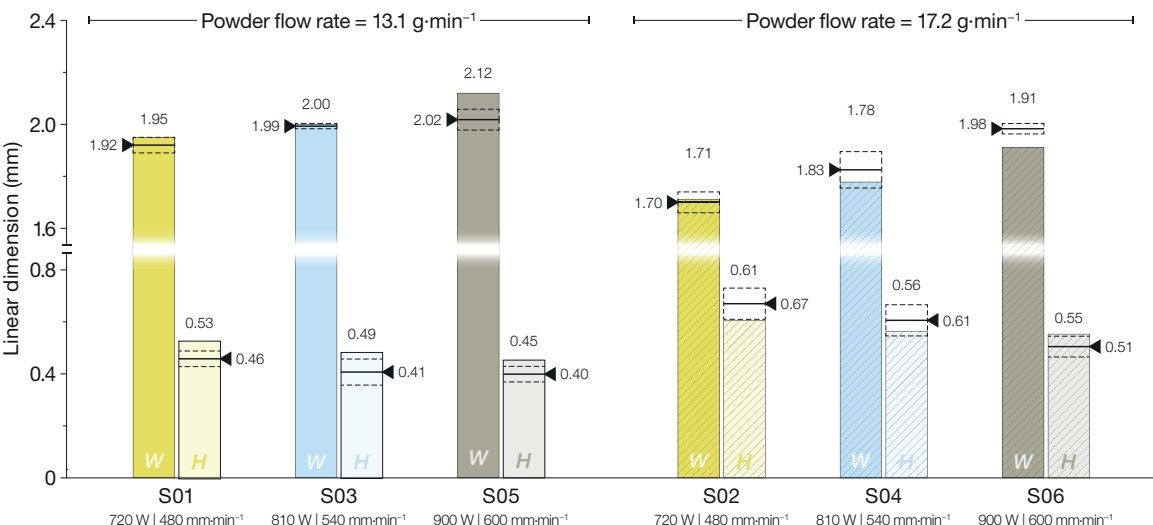

**Figure 11.** Widths and heights of the tracks as predicted by the grey-box model in comparison with the experimental data for squared tracks.

## 5. Conclusions

A grey-box model for the prediction of track characteristics and temperature field for the DED-LB/Powder process has been presented in this article. Initially, the MATLAB-implemented model was validated using rectilinear experimental tests with three distinct levels of travel speed and powder flow rate and two levels of power. The obtained results are closely consistent with the experimental data. In general, the results demonstrated that the size of the track is highly dependent on the selected power value in terms of both the energy delivered and the associated heat loss. Subsequently, the model was validated using experimental tests with more complex deposition paths, i.e., circular and squared tracks. At the same energy densities, the dimensions of the track changed with various values of travel speed, power and powder flow rate, with the power having a greater impact in this instance. Additionally, the laser path had an effect on the height of the track, resulting in taller tracks in the circular case. In conclusion, this model can be used to optimise the process parameters of the DED-LB/Powder process, which is a complex process with many factors and phenomena that influence its correct application. Potential future research

efforts might encompass a meticulous examination of the thermo-physical characteristics of the material. The incorporation of the Maragoni effect could enhance the accuracy of the proposed model. Furthermore, it is worth noting that the presence of steep temperature gradients leads to residual stresses in the components, causing deformations that ultimately result in dimensional and accuracy deviations. It is therefore reasonable to integrate this model into a thermomechanical formulation, thereby allowing the calculation of the stress state of the components and the associated deformation.

**Author Contributions:** Conceptualization, M.D.C. and A.S.; methodology, M.P., M.D.C. ans A.S.; formal analysis, M.P. and A.S.; investigation, M.P.; resources, E.A. and A.S.; writing—original draft preparation, M.P.; writing—review and editing, E.A. and A.S.; visualization, A.S. All authors have read and agreed to the published version of the manuscript.

**Funding:** The research was financial supported by Integrated Additive Manufacturing (IAM@PoliTO) at the Politecnico di Torino, Torino, Italy. This work has been partially supported by a "Ministero dell'Istruzione, dell'Università e della Ricerca" Award, "TESUN-83486178370409 finanziamento dipartimenti di eccellenza CAP. 1694 TIT. 232 ART. 6".

**Data Availability Statement:** Not applicable.

**Acknowledgments:** The authors would like to thank Eng. Lorenzo Alligo for contributions to the methodology.

**Conflicts of Interest:** The authors Mirna Poggi, Eleonora Atzeni and Alessandro Salmi declare no conflict of interest. The author Michele De Chirico is an employee of Prima Additive S.r.l. and the samples were produced using the LASERDYNE® 430 system by Prima Additive.

## Nomenclature

| | |
|---|---|
| $c_p$ | Specific heat capacity ($J \cdot kg^{-1} \cdot K^{-1}$) |
| $D$ | Melt pool depth (mm) |
| $f$ | Logic function |
| $\mathcal{G}$ | Green's function |
| $H$ | Deposition height of the track (mm) |
| $H'$ | Local deposition height (mm) |
| $h$ | Convective heat transfer coefficient ($W \cdot mm^{-2} \cdot K^{-1}$) |
| $L$ | Melt pool total length (mm) |
| $L_f$ | Melt pool front length (mm) |
| $L_r$ | Melt pool rear length (mm) |
| $L_m$ | Latent heat of fusion ($J \cdot kg^{-1}$) |
| $L_v$ | Latent heat of vaporization ($J \cdot kg^{-1}$) |
| $N$ | Number of segment of the laser path |
| $O$ | Coordinates system origin of $\mathfrak{R}$ |
| $O'$ | Coordinates system origin of $\mathfrak{R}'$ |
| $P$ | Laser power (W) |
| $P_{conv}$ | Power losses due to convection (W) |
| $P_{evap}$ | Power losses due to evaporation (W) |
| $P_{losses}$ | Overall power losses (W) |
| $P_{melt}$ | Power losses due to powder melting (W) |
| $P_{rad}$ | Power losses due to radiation (W) |
| $P_u$ | Useful power (W) |
| $Q_p$ | Powder flow rate ($g \cdot min^{-1}$) |
| $q_{pf}$ | Powder mass flux ($g \cdot s^{-1} \cdot mm^{-2}$) |
| $\mathfrak{R}$ | Substrate reference frame |
| $\mathfrak{R}'$ | Melt pool reference frame |
| $r$ | Laser beam radius (mm) |
| $r_{pf}$ | Powder stream radius (mm) |
| $s$ | Integration variable |
| $T$ | Temperature (K) |
| $T_a$ | Ambient temperature (K) |
| $T_{init}$ | Initial temperature (K) |

| | |
|---|---|
| $T_{melt}$ | Melting temperature (K) |
| $T_{mean}$ | Mean melt pool temperature (K) |
| $T_{peak}$ | Peak temperature (K) |
| $T^I$ | Temperature contribution due to the initial temperature (K) |
| $T^\Phi$ | Temperature contribution due to the heat flux (K) |
| $t$ | Time (s) |
| $V_a$ | Melt pool volume above the top of the substrate (mm$^3$) |
| $V_b$ | Melt pool volume below the top of the substrate (mm$^3$) |
| $v$ | Travel speed (mm·s$^{-1}$) |
| $W$ | Melt pool width (mm) |
| $(x, y, z)$ | Coordinate system of $\mathfrak{R}$ (mm) |
| $(x', y', z')$ | Coordinate system of $\mathfrak{R}'$ (mm) |
| $(x_c, y_c, z_c)$ | Coordinates of the center of laser beam in $\mathfrak{R}$ (mm) |
| $\alpha$ | Limit error of the loop cycle |
| $\beta$ | Substrate absorbivity |
| $\varepsilon$ | Emissivity |
| $\kappa$ | Thermal diffusivity (mm$^2$·s$^{-1}$) |
| $\lambda$ | Thermal conductivity (W·mm$^{-1}$·K$^{-1}$) |
| $\rho$ | Density (kg·mm$^{-3}$) |
| $\sigma$ | Stefan–Boltzman constant (W·mm$^{-2}$·K$^{-4}$) |
| $\tau$ | Integration variable |
| $\Phi$ | Heat flux (W·mm$^{-2}$) |
| $(\xi, \eta, \zeta)$ | Coordinates of a generic point in $\mathfrak{R}$ (mm) |
| $\bar{\phantom{x}}$ | Dimensionless variable |
| AM | Additive Manufacturing |
| DED | Directed Energy Deposition |
| DED-LB | Directed Energy Deposition using a laser based system |
| CFD | Computational Fluid Dynamics |
| FEM | Finite Element Method |
| PBF-LB | Powder Bed Fusion using a laser based system |

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
