# Peer review of "A Grey-Box Model of Laser Powder Directed Energy Deposition for Complex Scanning Strategy"

_metals, doi:10.3390/met13101763_

Round 1

Reviewer 1 Report

1. please add a list of abbreviations, symbols, and units

2. please highlight the advantages and disadvantages of the algorithm in relation to existing methods

3. if the authors base their work on current models (for example [22]), the novelty of the work should be highlighted and the advantages of generalizing a model present in the literature should be pointed out

4. practical application of the model should be highlighted

Author Response

Thank you very much for taking the time to review this manuscript. Please find the detailed responses in the attached file.

Reviewer 2 Report

The authors can briefly highlight the motivation/need for such a modeling approach in the abstract and also summarize the important outcomes/findings of the model predictions.

Mathematical expressions and equations are not clearly explained making it difficult to follow.

The mathematical formulations and assumptions made are not clearly and sufficiently justified. More detailed explanation is needed.

Impact of layer-to-layer and inter-track interactions during complex scanning strategies is not modeled in the current work. What will be the effect please explain.

No uncertainty/sensitivity analysis was done to establish confidence/limitations of model predictions.

The lack of discussion on further improvements/extensions needed makes the model incomplete.

Include more papers from 2023 in the introduction section to strengthen the work.

Explain the main limitations of the work.

No discussion on residual stresses is included in the work.

The effect of material properties is also not included in the work.

Lines 118 and 122 need more elaboration and explanation. Will it affect the accuracy of the results or not, if yes how much and why did the author make this assumption?

Author Response

(The authors gave the same response as above.)

Reviewer 3 Report

An interesting and well-prepared manuscript is presented. The authors prepared a grey-box model for the prediction of track characteristics and temperature field for the DED process. The model is explained in full details, and validated by several sets of experimental results. The model itself and the results could be very useful to the relevant research fields. The manuscript is encouraged to publish in the current form. 

Author Response

(The authors gave the same response as above.)

Reviewer 4 Report

This manuscript proposes a mixed numerical-analytical approach for predicting the geometric properties of the deposition track of the LP-DED surface. The manuscript is overall well written, while some revisions are needed:

1.      Line 101, the authors stated “In this research, the approach proposed by Ahsan and Pinkerton [22] was generalized,”. So, what does this mean? What is the contribution and innovation aspects of the authors work? The authors need to provide more explanations in this part to clearly state their contributions.

2.      The MATLAB code on Page 9 should be provided as the supplementary materials.

3.      For 4.1 Temperature distribution, are the results presented in this section all numerical simulation results? Why there are no experimental validation for temperature distribution?

4.      There are some grammatical errors throughout the manuscript and should be corrected.

The English is fine and some corrections are needed.

Author Response

(The authors gave the same response as above.)

Round 2

Reviewer 1 Report

Dear Authors,

The article has been improved as suggested. In my opinion, it can be considered for publication.

Best regards

Reviewer